# Concentrations of Cobalt, Chromium and Titanium and Immunological Changes after Primary Total Knee Arthroplasty—A Cohort Study with an 18-Year Follow-Up

**DOI:** 10.3390/jcm13040951

**Published:** 2024-02-07

**Authors:** Anders Brüggemann, Nils P. Hailer

**Affiliations:** Orthopaedics—Department of Surgical Sciences, Uppsala University, SE-751 85 Uppsala, Sweden; anders.bruggemann@uu.se

**Keywords:** total knee arthroplasty, cobalt, chromium, titanium, inductively coupled plasma–mass spectrometry, lymphocytes

## Abstract

**Background:** Total knee arthroplasty (TKA) generates elevated metal ion concentrations, but long-term changes in the concentrations of cobalt (Co), chromium (Cr) and titanium (Ti) after primary TKA and potential subsequent immune system activation—not limited to the joint but systemically—are not known. **Patients and Methods:** We conducted a cohort study on 26 patients with TKA (19 women; 16 with metal-backed and 10 with all-polyethylene tibial components) 18.3 years (min. 16.7, max. 20.5) after index TKA. A total of 69% of patients additionally underwent subsequent arthroplasty of the contralateral knee or either hip after the index surgery. Blood samples were analysed by inductively coupled plasma–mass spectrometry, and leukocytes were characterised by flow cytometry. Patients were clinically assessed using the Knee Society score and by plain radiography of the knee. **Results:** The median metal ion concentrations were 0.7 (0.1–13.0) µg/L for Co, 0.9 (0.4–5.0) µg/L for Cr, and 1.0 (0.2–13.0) µg/L for Ti. There was no relevant difference in systemic metal ion concentrations between patients exposed to single and multiple arthroplasties. The absolute count and proportion of CD3^+^CD4^+^CD8^+^ T cells was inversely correlated with both Co (rho −0.55, *p* = 0.003) and Cr concentrations (rho −0.59, *p* = 0.001). **Conclusions:** Between the first and second decades after primary TKA, in most patients, the concentrations of Co, Cr and Ti in blood samples were below the thresholds that are considered alarming. The negative correlation of Co and Cr concentrations with a subset of lymphocytes that commonly increases during immune activation is reassuring. This represents a worst-case scenario, underscoring that the investigated metal ions remain within reasonable ranges, even after additional hardware exposure.

## 1. Introduction

Patients with metal-on-metal hip arthroplasty can be asymptomatic despite elevated metal ion concentrations; thus, the active monitoring of exposed patients by conducting measurements of whole blood concentrations of cobalt (Co) and chromium (Cr) is recommended [1]. Several national guidelines, including the Swedish ones, define thresholds between 2 and 3 μg/L as the upper limit for these ions [2]. Implants designed for primary total knee arthroplasty (TKA) usually have no metal-on-metal articulation, and guidelines for the follow-up of TKA do not regularly include the measurements of metal ion concentrations. However, the classification of Co as potentially carcinogenic by the European Union has instigated renewed interest in the possible consequences of exposure to Co in TKA patients [3].

Elevated metal ion concentrations are described in the early years after primary TKA. Co and Cr concentrations of 0.3 μg/L and 0.6 μg/L were found one year after the insertion of standard metal-backed TKA, with no Co measurement above 1 μg/L [4]. In a different study including baseline data, Co concentrations are described to be unchanged when compared with pre-operative measurements one year after conventional TKA [5]. In the short term, the reported concentrations after primary TKA are, thus, far below those found after hip resurfacing, and below the safety thresholds that have been defined in the context of metal-on-metal hip arthroplasty.

No long-term investigation of metal ion concentrations after TKA and no analysis of immune system alterations after the insertion of a TKA is known to us. The present study therefore aims to (1) analyse the blood concentrations of Co, Cr and Ti, the three major metals constituting alloys commonly used in TKA designs, at least one decade after index surgery, and (2) to assess whether changes in lymphocyte subsets as a sign of any systemic immunological activation are correlated with blood metal ion concentrations.

## 2. Patients and Methods

### 2.1. Study Design and Participants

This is an observational cohort study on patients who underwent a cemented TKA between the years 1994 and 1997 and who were included in three different randomised studies. The first cohort (40 patients) received a Freeman–Samuelson (FS) knee prosthesis with either all-polyethylene (*n* = 20) or metal-backed (*n* = 20) tibial components, all cemented with Palacos (Heraeus, Hanau, Germany). The second cohort (40 patients) received an AGC (Anatomical Graduated Component) knee prosthesis with either all-polyethylene (*n* = 20) or metal-backed (*n* = 20) tibial components, using Palacos bone cement. The remaining 51 patients (the third cohort) received an AMK (Anatomic Modular Knee) prosthesis with a metal-backed tibial component; in this subgroup, half the prostheses were cemented with CMW-1 (DePuy Synthes, Raynham, MA, USA, *n* = 26) and the other half with Palacos (*n* = 25). The primary outcome of all studies was implant migration, as measured by radiostereometry. At the time of this long-term follow-up, 52 patients were still alive, of whom a total of 26 patients were available for clinical examination; the remaining 26 declined to participate or were unfit to attend outpatient visits. Thus, of the twenty-six patients attending follow-up, nine patients had received an AGC TKA (four all-polyethylene, five metal-backed tibial components), ten patients had received an FS TKA (six all-polyethylene, four metal-backed tibial components), and the remaining seven patients had received the AMK TKA (only metal-backed tibial components).

### 2.2. Metal Ion Analysis

Blood samples were collected in two 10 mL EDTA anti-coagulated trace-metal-free tubes by a registered nurse using an 18-gauge intravenous cannula, with the central stainless-steel needle removed. The first tube was used for immunological analysis, and the second for determining the metal ion concentration in order to reduce the potential for metal ion contamination, but no further measures were undertaken to control for contamination. The vial for the metal analysis ion was sent to an accredited external laboratory specialising in trace metal analysis (ALS AB, Luleå, Sweden). The samples were dissolved in HNO_3_ in a microwave oven. They were then analysed for Cr, Co and Ti using inductively coupled plasma—mass spectrometry (ICP-MS) according to SS EN ISO17294-1, 2. The detection limits for the individual elements were 0.05 µg/L for Co, 0.2 µg/L for Cr, and 0.05 µg/L for titanium, all measurements below the detection limit were set equal to the detection limit, and all ion concentrations are presented in μg/L.

### 2.3. Immunological Analysis

Immunological analysis was performed by flow cytometry at Uppsala University Hospital using commercially available antibodies (CD3, CD8, CD45, CD4: Tetrachrome Beckman Coulter, Bromma, Sweden, cat. No. 6607013; CD3, CD 16/56, CD45, CD19: Tetrachrome, cat. No. 332779; HLA DR: Becton Dickinson, cat. No. 6604366; CD4: Becton-Dickinson, cat. No. 345768; CD8: Dako, cat. No. C7079). Results are given as numbers per nL, as percentages or as ratios where applicable.

### 2.4. Clinical and Radiographic Assessment

All analysed patients were evaluated according to the clinical scoring systems of the Knee Society [6]. This assessment consists of two parts: the knee score assessing knee function with respect to pain, stability and range of motion, and the function score assessing maximum walking distance, the ability to climb stairs and the need for walking aids. Plain radiography of the knee with anterior–posterior and lateral views was performed, and these were assessed for aseptic loosening. These evaluations were independently performed by two observers (A.G. and N.P.H.).

### 2.5. Ethics

Uppsala Ethics Committee approved both the three original randomised trials that the study participants were originally included in (approval number 104/93) and the long-term follow-up of the cohort described here (approval number 2012/013, date of approval: 14 May 2012). An additional approval was requested and granted by the Swedish Ethical Review Authority to perform a medical chart review of all included patients in order to ascertain the presence or absence of additional TKA or total hip arthroplasty (approval number: 2022-00905-02, date of approval: 2 March 2022). The research described here followed the updated version of the Helsinki declaration.

### 2.6. Statistical Analysis

Data distributions were visualised using histograms and analysed for normality through visual inspection of the histograms and density functions. Metal ion measurements, lymphocyte counts and proportions, and knee scores deviated considerably from normal distributions; therefore, medians, ranges and interquartile ranges (IQR) were used to describe these data. The non-parametric Wilcoxon–Mann–Whitney U-test was used to compare groups, and Spearman’s rho statistic was calculated to investigate correlations. R version 3.6.1 statistical software was used for analysis, *p*-values < 0.05 were considered indicative of statistical significance, and due to the exploratory nature of our study, no multiplicity correction was performed.

## 3. Results

### 3.1. Characteristics of the Study Population

The study population consisted of 26 patients (19 women) with a mean age at surgery of 65.3 years (SD 5.3; Table 1). This cohort of 26 patients was derived from two separate randomised controlled trials that compared metal-backed with all-polyethylene designs (either of the AGC or FS brand), or from a third randomised controlled trial from the same decade that compared two types of cement used in TKA surgery (CMW-1 with Palacos; see Section 2). The presence of both metal-backed (*n* = 16) and all-polyethylene tibial components (*n* = 10) permitted us to perform subgroup analyses on patients, comparing these two different tibial component designs with respect to metal ion concentrations. Eighteen patients (69%) had received either a contralateral TKA (*n* = 2) or a conventional total hip arthroplasty (THA) with a metal head articulating with a polyethylene liner.

### 3.2. Metal Ion Concentrations

The median Co concentration in the entire cohort was 0.7 (IQR: 0.89, range: 0.1–13.0) µg/L. When divided by the type of tibial component, the median Co concentrations were slightly lower among patients with all-polyethylene than among those with metal-backed tibial components, but the difference was not statistically significant (Table 2).

The median Cr concentration in the entire cohort was 0.9 (IQR: 0.85, range: 0.4–5) µg/L. As for Co concentrations, the median Cr concentrations were slightly but not statistically significantly lower among patients with all-polyethylene than among those with metal-backed tibial components (Table 2).

The median concentration of Ti in the pooled cohort was 1 (IQR: 0.63, range: 0.2–13) µg/L. The median Ti concentration in patients with all-polyethylene tibial components was slightly lower than in those with metal-backed, although the maximal Ti concentration (13 µg/L) was found in a patient with an all-polyethylene component. There was no statistically significant difference in Ti concentrations between these two groups (Table 2).

The concentrations of Co and Cr correlated strongly with each other (rho 0.69, *p* < 0.001). In contrast, we observed no correlations between Co and Ti (rho −0.073, *p* = 0.7) or between Cr and Ti (rho 0.1, *p* = 0.6).

Since the majority of patients had received additional arthroplasties in other joints, we compared metal ion concentrations in those with singular and those with multiple arthroplasties, but found no statistically significant differences for any of the investigated ions. However, the maximum Co and Ti concentrations were found in the group of patients with multiple arthroplasties (Table 3).

### 3.3. Immunological Analyses

The counts and relative proportions of the measured lymphocyte subsets were within the normal ranges, according to local reference values (Table 4). When analysing potential correlations between lymphocyte counts or concentrations and metal ion concentrations, we found a statistically significant, negative correlation between the absolute count and the proportion of CD3^+^CD4^+^CD8^+^ cells with Co concentration (rho −0.55, *p* = 0.0034 for the absolute count). Similarly, we observed a negative correlation of the absolute count and the proportion of CD3^+^CD4^+^CD8^+^ cells with Cr concentrations (rho −0.59, *p* = 0.0014 for the absolute count). There were some other statistically significant negative correlations between lymphocyte subsets, on the one hand, and Co concentrations on the other hand (Table 3). We then investigated whether the subsets of lymphocytes differed between patients with all-polyethylene and metal-backed tibial components, and found some with statistically significant differences between groups (absolute leukocyte count, CD3^+^, CD3^+^CD4^+^, CD3^+^CD4^+^CD8^+^, CD45^+^, DR/CD4^+^; Table 5).

### 3.4. Clinical and Radiographic Findings

The median knee score was 83.5 (50–100; IQR 12) and the median function score was 75.0 (0–100; IQR 45). When divided by the type of tibial component, no statistically significant differences between groups were seen. Radiographic signs of loosening in the form of radiolucent lines under the tibial component were found in only one patient, but this patient was asymptomatic.

## 4. Discussion

### 4.1. Principal Findings

Our main finding is that Co and Cr concentrations in the blood are slightly—yet not harmfully—elevated almost two decades after TKA, regardless of exposure to subsequent arthroplasties in the hip or knee. With very few exceptions, the concentrations of these ions are well below the thresholds that have been defined as potentially dangerous in the context of metal-on-metal hip arthroplasty. Ti concentrations are also slightly elevated, but lower than the concentrations that are found a decade after cementless total hip arthroplasty. No signs of immune cell activation are observed in our cohort.

### 4.2. Agreement and Disagreement with Other Studies

The increased concentration of Co and Cr in patients with metal-on-metal articulations in the short term is a known phenomenon [7]. Even a decade after metal-on-metal hip arthroplasty, Co concentrations seem to remain unchanged, whereas Cr concentrations may slightly decline [8]. In some metal-on-metal hip devices, however, Co also declines after ten years [9]. By definition, our cross-sectional study design provides no longitudinal data, but extrapolation from studies on metal ion concentrations after TKA measured at short-term follow-up [4,5] indicates that there is no gross accumulation of metal ions in the long term after TKA.

The pathogenesis of elevated metal ion concentrations after TKA is a combination of corrosion and wear, as found in a study of about 50 CoCr-alloy femoral TKA components retrieved after a mean period of 18 years after index surgery [10]. Almost all investigated implants from five major manufacturers showed signs of abrasive third body wear, and hence, most metal ion release can be considered to be due to third-body wear, but other mechanisms such as electrolysis may contribute as well.

Some hinged TKA designs that are used in revision surgery have a hinge mechanism that includes a metal-on-metal bearing. Co and Cr concentrations far above the commonly agreed threshold of 2 μg/L are described around two years after the use of certain hinged TKA, with maximal values up to 69 μg/L [11]. In a different study on hinged TKA, Co concentrations above 5 μg/L are reported to be present in the majority of investigated patients at a mean follow-up of three years, and Cr levels are also elevated [12].

The development of local or systemic Ti concentrations after TKA is much less investigated, although contemporary TKA designs contain Ti in the tibial baseplate. In studies both on cementless and cemented TKA, median Ti concentrations around 1 μg/L are reported one year after index surgery [4,13]. This can be compared to one of very few studies following Ti concentrations after total hip arthroplasty, where peak values of almost 3 µg/L are described three years after index surgery [14]. Another study on a small cohort of patients with various Ti implants, but no TKA, describes Ti concentrations ranging from 2 to 6 µg/L [15]. Exceptionally high Ti concentrations in the context of TKA are only described in a study including a small subgroup of patients with failed patellar components [16]. The median Ti concentrations found in our cohort are thus at the lower end of the previously described concentration ranges. Of note, compared to Co or Cr, Ti is generally considered relatively inert, but in the context of total hip arthroplasty with CoCr modular neck junctions and bearings other than metal-on-metal, Ti particles seem to be colocalised with Co and Cr nanoparticles [17]. Importantly, the use of all-polyethylene tibial components in TKA avoids exposure to Ti, and this concept has proven to be fairly successful, as indicated by a large Swedish arthroplasty register study [18].

An adverse reaction to metal debris (ARMD) is a common phenomenon after hip resurfacing or total hip arthroplasty using large metal-on-metal articulations. Elevated metal ion concentrations can affect the immune system, which may explain some of the local, peri-articular phenomena observed in the context of metal-on-metal devices in the hip, including the development of ARMD and pseudotumours [19,20]. ARMD associated with the corrosion of modular TKA has been reported [21], and the development of ARMD or even local pseudotumours has been described in a large series of revision TKAs [22]. However, these authors did not report on potential systemic immune reactions related to the development of peri-articular ARMD.

As in the occurrence of ARMD after hip resurfacing, the basic pathophysiological principle of systemic immune activation should be applicable even after TKA, and our study seems to be the first to investigate this topic of systemic immune cell activation after TKA. We fail to detect said activation, yet observe some statistically significant differences in lymphocyte subpopulations between patients with all-polyethylene and metal-backed tibial components. Given the multiplicity of tests performed in this study, this finding may represent a Type-I error. Considering the relatively small numbers of patients in each subgroup, we refrain from speculating as to whether the presence or absence of Ti may have affected lymphocyte subsets. Nonetheless, the finding is interesting and merits further investigations on larger cohorts. 

The classification of Co as potentially carcinogenic to humans that was issued by the European commission in 2021 has raised world-wide concerns related to the risk of developing cancer after exposure to orthopaedic implants. However, large-scale register studies indicate no increased overall risk of cancer in patients exposed to total hip arthroplasty [23,24,25], and meta-analyses including occupational exposure to Co also fail to uncover increased risks of cancer [26]. While this is reassuring to both patients with implanted hip and knee replacements and their care givers, it does not exclude the possibility of effects on the immunological or nervous system in vivo. Hence, studies such as the present work are of importance to investigate any possible association between elevated metal ion concentrations and systemic effects.

### 4.3. Strengths and Weaknesses of This Study

Our study is limited by numerous weaknesses. First of all, measurements of metal ion concentrations and immunological alterations were not part of the primary or secondary outcome measures of the original trial designs, and the sample sizes were consequently not adequate to investigate these research questions. We also lack baseline data on metal ion concentrations, further limiting our conclusions. The availability of only 26 of the original 116 patients raises concerns related to selection bias, and the deceased patients and those not willing or able to attend clinical follow-up who were excluded from our analyses might have had the highest metal ion concentrations. The absence of biopsies, and hence, the inability to perform histological examinations in our study, prevents us from drawing conclusions on the absence or presence of a lymphocytic infiltrate in the local tissue reaction to wear debris.

The presence of multiple arthroplasties in about two thirds of our patients is a considerable limitation. Nonetheless, with a minimum follow-up of 17 years after the index TKA, this is not entirely surprising when considering the systemic character of osteoarthritis [27]. Only two of the patients in our cohort were exposed to another TKA, and the remaining 16 had THA implants; however, they were only metal-on-polyethylene articulations. We recognise that this may distort our metal ion measurements and the correlated changes in lymphocyte subsets. Yet, we consider the influence of additional hip implants to be of minor influence. Co and Cr concentrations in patients with metal-on-polyethylene THA are low 16 years after THA, with Co measured at 0.4 µg/L and Cr at 1.0 µg/L [28]. One may also argue that our findings represent a worst-case scenario: two thirds of our patients were exposed to an arthroplasty, mostly THA, in addition to the TKA that was our primary interest. Nonetheless, their metal ion concentrations are far from the range of those found in patients exposed to metal-on-metal THA or hinged revision TKA.

Our assessment of Ti concentrations using ICP-MS rather than inductively coupled plasma–optical emission spectroscopy (ICP-OES) is in accordance with current recommendations [29]. Our data description uses medians and statistical inference is based on non-parametric methods, which also complies with recommendations on the interpretation of Ti measurements that are referred to above.

An important strength of our study is the long-term follow-up of both metal ion concentrations and immunological parameters in a well-defined population of primary TKA patients. None of the cited studies on the topic of metal ion exposure after TKA addresses this issue that is well characterised after THA, both in patients with conventional metal-on-polyethylene articulations and in those with metal-on-metal devices.

## 5. Conclusions

Our study indicates that patients who have been exposed to TKA for almost two decades have low blood concentrations of Co, Cr and Ti, even when exposed to additional arthroplasties in the hip or knee joint. This is an important finding since the classification of Co as a potentially cancerogenic substance has raised concerns, both among patients and orthopaedic surgeons, related to potential cancer risks after TKA. Secondly, a note of caution remains around a few patients with Co and Cr concentrations above the reference values defined for hip arthroplasty patients. Thirdly, patients in our cohort fail to show signs of immunological activation that have been previously described in the context of metal ion exposure after hip arthroplasty, which is also a reassuring finding.

## Figures and Tables

**Table 1 jcm-13-00951-t001:** Characteristics of the study population, first divided by those who received all-polyethylene or metal-backed tibial components, and then, summarised (Overall). Numbers indicate *n* and percentages (in parentheses) unless otherwise stated.

	All-Poly (*N* = 10)	Metal-Backed (*N* = 16)	Overall (*N* = 26)
Sex			
Male	1 (10.0%)	6 (37.5%)	7 (26.9%)
Female	9 (90.0%)	10 (62.5%)	19 (73.1%)
Side			
Right	5 (50.0%)	12 (75.0%)	17 (65.4%)
Left	5 (50.0%)	4 (25.0%)	9 (34.6%)
Age at Surgery			
Mean (range)	64 (53–73)	66 (57–75)	65 (53–75)
Follow up [years]			
Mean (range)	18 (17–19)	18 (17–21)	18 (17–21)

**Table 2 jcm-13-00951-t002:** Metal ion measurements performed by ICP-MS (all concentrations in µg/L). Measurements are first divided by patients who received all-polyethylene or metal-backed tibial components, and then, summarised (Overall). * Standard deviation.

	All-Poly (*N* = 10)	Metal-Backed (*N* = 16)	Overall (*N* = 26)
Cobalt			
Median (IQR *; range)	0.56 (0.35; 0.13–3.2)	0.89 (1.1; 0.14–13)	0.71 (0.89; 0.13–13)
Chrome			
Median (IQR; range)	0.62 (0.51; 0.37–1.9)	1.0 (1.0; 0.37–5.0)	0.91 (0.85; 0.37–5.0)
Titanium			
Median (IQR; range)	0.42 (0.61; 0.16–13)	1.0 (0.38; 0.22–1.7)	1.0 (0.63; 0.16–13)

**Table 3 jcm-13-00951-t003:** Metal ion measurements performed by ICP-MS (all concentrations in µg/L), comparing metal ion concentrations in patients with single vs. multiple arthroplasties. * Total knee arthroplasty; ** interquartile range.

	Singular TKA * (*N* = 8)	Multiple Arthroplasties (*N* = 18)	Overall (*N* = 26)
Cobalt			
Median (IQR **; range)	1.1 (0.49; 0.53–2.7)	0.59 (0.50; 0.13–13)	0.71 (0.89; 0.13–13)
Chrome			
Median (IQR; range)	1.3 (0.60; 0.44–5.0)	0.62 (0.53; 0.37–3.9)	0.91 (0.85; 0.37–5.0)
Titanium			
Median (IQR; range)	0.46 (0.72; 0.22–1.2)	1.0 (0.54; 0.16–13)	1.0 (0.63; 0.16–13)

**Table 4 jcm-13-00951-t004:** Lymphocyte counts per nL, or percentages (%), or ratios, given as medians with ranges, and correlations of these values with cobalt, chromium and titanium concentrations. Rho and *p*-values are derived from Spearman’s rank correlation.

				Correlations					
				Cobalt		Chrome		Titanium	
	Median	Min.	Max.	Rho	*p*	Rho	*p*	Rho	*p*
Leukocyte count	7.65	4.8	13	−0.18	0.37	−0.26	0.2	−0.32	0.11
CD3 (%)	72.65	41	83	−0.04	0.85	−0.05	0.8	−0.12	0.56
CD3	1099.5	390	1906	−0.36	0.07	−0.23	0.25	−0.3	0.14
CD3CD8 (%)	18.75	5.1	38.1	0.15	0.47	0.08	0.7	−0.08	0.68
CD3CD8	341.5	47.3	716	−0.07	0.73	−0.07	0.73	−0.07	0.73
CD3CD4 (%)	51.55	25.9	67.3	−0.31	0.12	−0.26	0.2	−0.09	0.66
CD3CD4	782.5	246	1446	−0.47	0.02	−0.32	0.11	−0.29	0.15
CD3CD4CD8 (%)	1.015	0.18	4.6	−0.5	0.01	−0.59	0.002	−0.15	0.48
CD3CD4CD8	15.55	1.7	103	−0.55	0.003	−0.59	0.001	−0.16	0.44
CD16CD56 (%)	16.85	7.3	42.3	0.14	0.49	0.14	0.49	−0.08	0.68
CD16CD56	272.5	112	609	−0.05	0.82	0.07	0.72	−0.14	0.49
CD19 (%)	9.1	2.2	23	−0.08	0.68	−0.12	0.56	0.24	0.24
CD19	165.5	28	305	−0.33	0.1	−0.28	0.17	0.06	0.77
CD45	1496	937	2716	−0.39	0.05	−0.25	0.23	−0.28	0.17
Ratio CD4/CD8	2.77	0.95	11.13	−0.24	0.23	−0.17	0.41	−0.04	0.84
DR/CD4 (%)	3.8	1.7	24	0.45	0.03	0.37	0.07	0.37	0.07
DR/CD8 (%)	1.4	0.1	14	0.21	0.32	0.14	0.5	0.31	0.14

**Table 5 jcm-13-00951-t005:** Differences in lymphocyte populations between patients who received all-polyethylene or metal-backed tibial components. Mann–Whitney *U* statistics and two-tailed *p*-values derived from Wilcoxon–Mann–Whitney rank sum test.

Variable	Mann–Whitney *U* Statistic	*p*
Leukocyte count	122	0.03
CD3 (%)	106	0.18
CD3	126	0.01
CD3CD8 (%)	99.5	0.3
CD3CD8	111	0.1
CD3CD4 (%)	98.5	0.34
CD3CD4	121	0.03
CD3CD4CD8 (%)	114	0.07
CD3CD4CD8	124	0.02
CD16CD56 (%)	73	0.73
CD16CD56	102	0.26
CD19 (%)	54	0.18
CD19	78	0.94
CD45	124	0.02
Ratio CD4/CD8	71	0.66
DR/CD4 (%)	19	0.002
DR/CD8 (%)	52	0.2

## Data Availability

Due to Swedish legislation data on individual patients may not be shared. Upon reasonable request and approval by the Ethical Review Authority aggregated data may be shared.

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
