# Peer review of "Concentrations of Cobalt, Chromium and Titanium and Immunological Changes after Primary Total Knee Arthroplasty—A Cohort Study with an 18-Year Follow-Up"

_jcm, 2024, doi:10.3390/jcm13040951_

Round 1
Reviewer 1 Report
Comments and Suggestions for Authors
The Authors examine the blood concentrations of Co, Cr, Ti and presence of lymphocytes in a cohort of patients implanted long-term with total knee arthroplasty (TKA). The following comments have the intent of strengthen the manuscript and increase the interest of the readership.
Patients and Methods
Page 3, lines 90-95: Line No explanation is provided regarding the validation of the lymphocytic count in blood compared to the periprosthetic neo-synovium or synovial fluid. This is a very important question which needs to be answered since it can invalidate the entire study. There is no citation provided regarding previous studies using this method for the stated purposes of the study and a brief review of the literature results in the value of the neutrophil to lymphocyte ratio blood count to predict periprosthetic joint infection and not for evaluating possible mechanisms of metal hypersensitivity/allergy, usually measured by histopathological analysis of periprosthetic soft tissue collected at implant revision surgery.
Discusssion
Page 6, lines 208-210: Apart from well documented mechanically assisted crevice corrosion occurring at modular femoral junction, inflammatory cell induced corrosion is still a controversial mechanism with no demonstration of actual macrophage adhesion and corrosion in vivo. Therefore, in the great majority of the cases, metal wear of CoCr metallic femoral condyle is generated by third body wear, unless complete degradation of the plastic insert causing metal-on metal contact.
Page 6, lines 234-235: Studies reporting adverse local tissue reactions in TKA have been described, in particular secondary metallic junction corrosion [Arnholt C, Donald DW, Tohfafarosh M, Gilbert JL, Rimnac CL, Kurtz SM, the Implant Research Writing Committee, Klein G, Mont MA, Parvizi J, Cates HE, Lee GC, Malkani A, Kraay M. Mechanically assisted taper corrosion in modular TKA. J Arthroplasty 2014;29 Suppl 9:205–08; Christiner T, Pabbruwe MB, Kop AM, Parry J, Clark G, Collopy, D. Taper Corrosion and Adverse Local Tissue Reactions in Patients with a Modular Knee Prosthesis. JBJS Open Access 2018;1. doi: 10.2106/jbjs.oa.18.00019; Kurmis, A. P., Herman, A., McIntyre, A. R., Masri, B. A., & Garbuz, D. S. (2019). Pseudotumors and High-Grade Aseptic Lymphocyte-Dominated Vasculitis-Associated Lesions Around Total Knee Replacements Identified at Aseptic Revision Surgery: Findings of a Large-Scale Histologic Review. The Journal of arthroplasty, 34(10), 2434–2438.]
Page 6, lines 235-236: It is questionable if the methodology used can exclude any immunological reaction in the periprosthetic soft tissue, although lymphocytic predominant adverse local tissue reactions occur at early time of implantation, unless the corrosion occurs much later.
Page 6, lines 239-242: Ti is generally considered inert compared to Co and Cr, unless particles/ions are localized or co-localized with nanoparticle aggregates of CoCr Mo [Xia Z, Ricciardi BF, Liu Z, von Ruhland C, Ward M, Lord A, et al. Nano-analyses of wear particles from metal-on-metal and non-metal-on-metal dual modular neck hip arthroplasty. Nanomedicine 2017;13: 1205-217].
Page 6, lines 243-248: If the lack of increased risk for cancer may be true, risks for the immunological system and nervous system might be underestimated and should be object of rigorous studies.
Reviewer 2 Report
Comments and Suggestions for Authors
Concentrations of cobalt, chromium and titanium and immunological changes after primary total knee arthroplasty—A cohort study with 18.3 years follow-up
The authors aimed to assess the long-term changes in concentrations of cobalt (Co), chromium (Cr) and titanium (Ti) and any changes in lymphocyte subsets (a sign of immunological activation) in a cohort of 26 patients (19 women) who had undergone primary total knee arthroplasty (TKA) at a mean of 18.3 years earlier (range 16.7-20.5 years).16 patients had metal-backed and 10 all-polyethylene tibial components.
Blood samples were analysed by inductively coupled plasma–mass spectrometry and leukocytes were characterized by flow cytometry. Patients were also clinically assessed using the Knee society scores and by plain knee radiography.
The authors found that the median metal ion concentrations were 0.7 (0.1-13.0) µg/L for Co, 0.9 (0.4-5.0) µg/L for Cr, and 1.0 (0.2-13.0) µg/L for Ti; elevated, but not dangerously so.
Eighteen patients (69%) had received either a contralateral TKA (n=2) or a conventional total hip arthroplasty (THA) with a metal head articulating with a polyethylene liner. There was no relevant difference in metal ion concentrations between patients who had one and multiple arthroplasties; however, the patients with the maximum Co and Ti concentrations were found in the group who had multiple arthroplasties
The absolute count and proportion of CD3+CD4+CD8+ T cells was (reassuringly) inversely correlated with both Co (rho -0.55, p=0.0034) and Cr concentrations (rho -0.59, p=0.0014). the context of 289 metal ion exposure after hip arthroplasty.
The authors concluded that these patients (almost 2 decades post TKA) had concentrations of Co, Cr and Ti below the thresholds that are considered dangerous; even in those patients with additional hardware. Their data also reassuringly demonstrated that there was a negative correlation of Co and Cr concentrations with the subset of lymphocytes that commonly increases during immune activation.
The authors are quite candid about the shortcomings of the study. Nevertheless the paper does add to the literature and I am happy to support its publication. The paper is structured well, and the methodology and discussion is good.
Please make this line clearer in the abstract. “69% of patients had received additional arthroplasties after index surgery”. I initially read this as 69% of these patients had undergone further arthroplasty (ie: revision TKA) surgery.
Maybe change to “69% of these patients had undergone arthroplasty of the contralateral knee or hip after index surgery”.
Line 78. Though the blood tubes were metal ion-free, then cannula would have been inserted with a metal needle. Have the authors controlled for this?
Some grammatical and language corrections are needed.
Comments on the Quality of English LanguageConcentrations of cobalt, chromium and titanium and immunological changes after primary total knee arthroplasty—A cohort study with 18.3 years follow-up
The authors aimed to assess the long-term changes in concentrations of cobalt (Co), chromium (Cr) and titanium (Ti) and any changes in lymphocyte subsets (a sign of immunological activation) in a cohort of 26 patients (19 women) who had undergone primary total knee arthroplasty (TKA) at a mean of 18.3 years earlier (range 16.7-20.5 years).16 patients had metal-backed and 10 all-polyethylene tibial components.
Blood samples were analysed by inductively coupled plasma–mass spectrometry and leukocytes were characterized by flow cytometry. Patients were also clinically assessed using the Knee society scores and by plain knee radiography.
The authors found that the median metal ion concentrations were 0.7 (0.1-13.0) µg/L for Co, 0.9 (0.4-5.0) µg/L for Cr, and 1.0 (0.2-13.0) µg/L for Ti; elevated, but not dangerously so.
Eighteen patients (69%) had received either a contralateral TKA (n=2) or a conventional total hip arthroplasty (THA) with a metal head articulating with a polyethylene liner. There was no relevant difference in metal ion concentrations between patients who had one and multiple arthroplasties; however, the patients with the maximum Co and Ti concentrations were found in the group who had multiple arthroplasties
The absolute count and proportion of CD3+CD4+CD8+ T cells was (reassuringly) inversely correlated with both Co (rho -0.55, p=0.0034) and Cr concentrations (rho -0.59, p=0.0014). the context of 289 metal ion exposure after hip arthroplasty.
The authors concluded that these patients (almost 2 decades post TKA) had concentrations of Co, Cr and Ti below the thresholds that are considered dangerous; even in those patients with additional hardware. Their data also reassuringly demonstrated that there was a negative correlation of Co and Cr concentrations with the subset of lymphocytes that commonly increases during immune activation.
The authors are quite candid about the shortcomings of the study. Nevertheless the paper does add to the literature and I am happy to support its publication. The paper is structured well, and the methodology and discussion is good.
Please make this line clearer in the abstract. “69% of patients had received additional arthroplasties after index surgery”. I initially read this as 69% of these patients had undergone further arthroplasty (ie: revision TKA) surgery.
Maybe change to “69% of these patients had undergone arthroplasty of the contralateral knee or hip after index surgery”.
Line 78. Though the blood tubes were metal ion-free, then cannula would have been inserted with a metal needle. Have the authors controlled for this?
Some grammatical and language corrections are needed.
Round 2
Reviewer 1 Report
Comments and Suggestions for Authors
Page 10, line 219: And hence, most metal ion release can be considered to be due to 3rd body wear, but other mechanisms such as oxidative stress electrolysis do contribute as well. Suggested change: may contribute as well instead of do since you might have variation according to implant configuration. You need to take into account that the TKA you studied might not be on the market anymore and that the findings cannot be generalized to other TKA implants too.
Page 10, line 238: To note, compared to Co or Cr, Ti is generally considered relatively inert, but in the context of hip resurfacing, Ti particles seem to be colocalized with Co and Cr nanoparticles. The most toxic is Co, Cr much less than Co, and Ti relatively inert. It cannot be in the context of MoM resurfacing, because Ti particles are not present unless there is impingement of the femoral cup against the acetabular rim and were not found in the cases examined in the study. The cited study showed presence of Ti particles in MoM THA and colocalization in Non MoM THA with CoCr dual modular neck with corrosion at the neck-stem junction. Please correct the statement.
Page 8, line 282: The absence of biopsies, and hence the inability to perform histological examinations in our study, prevents us from drawing conclusions on the development of pseudotumours. The presence of "pseudotumors" cannot be assessed on synovial biopsies. Please modify the sentence as follows: on the absence or presence of a lymphocytic infiltrate in the local tissue reaction to wear debris.
Author Response
Reviewer 1:
Page 10, line 219: And hence, most metal ion release can be considered to be due to 3rd body wear, but other mechanisms such as oxidative stress electrolysis do contribute as well. Suggested change: may contribute as well instead of do since you might have variation according to implant configuration. You need to take into account that the TKA you studied might not be on the market anymore and that the findings cannot be generalized to other TKA implants too.
We agree with the reviewer and changed the manuscript accordingly.
Page 10, line 238: To note, compared to Co or Cr, Ti is generally considered relatively inert, but in the context of hip resurfacing, Ti particles seem to be colocalized with Co and Cr nanoparticles. The most toxic is Co, Cr much less than Co, and Ti relatively inert. It cannot be in the context of MoM resurfacing, because Ti particles are not present unless there is impingement of the femoral cup against the acetabular rim and were not found in the cases examined in the study. The cited study showed presence of Ti particles in MoM THA and colocalization in Non MoM THA with CoCr dual modular neck with corrosion at the neck-stem junction. Please correct the statement.
We deeply apologize for this mistake, we corrected the statement as requested to better reflect the cited study.
Page 8, line 282: The absence of biopsies, and hence the inability to perform histological examinations in our study, prevents us from drawing conclusions on the development of pseudotumours. The presence of "pseudotumors" cannot be assessed on synovial biopsies. Please modify the sentence as follows: on the absence or presence of a lymphocytic infiltrate in the local tissue reaction to wear debris.
Once again, a good point by the reviewer. We changed the manuscript as requested.